# Offline Robot Reinforcement Learning with Uncertainty-Guided Human Expert Sampling

**Ashish Kumar**
Offworld Inc., United States
ashish.kumar@offworld.ai

**Ilya Kuzovkin**
Offworld Inc., United States
ilya.kuzovkin@offworld.ai

## Abstract

Recent advances in batch (offline) reinforcement learning have shown promising results in learning from available offline data and proved offline reinforcement learning to be an essential toolkit in learning control policies in a model-free setting. An offline reinforcement learning algorithm applied to a dataset collected by a suboptimal non-learning-based algorithm can result in a policy that outperforms the behavior agent used to collect the data. Such a scenario is frequent in robotics, where existing automation is collecting operational data. Although offline learning techniques can learn from data generated by a sub-optimal behavior agent, there is still an opportunity to improve the sample complexity of existing offline reinforcement learning algorithms by strategically introducing human demonstration data into the training process. To this end, we propose a novel approach that uses uncertainty estimation to trigger the injection of human demonstration data and guide policy training towards optimal behavior while reducing overall sample complexity. Our experiments show that this approach is more sample efficient when compared to a naive way of combining expert data with data collected from a sub-optimal agent. We augmented an existing offline reinforcement learning algorithm Conservative Q-Learning with our approach and performed experiments on data collected from MuJoCo and OffWorld Gym learning environments.

## 1 Introduction

The field of *offline reinforcement learning* has emerged in the past few years as a way of training reinforcement learning (RL) agents from previously collected experiences and without the need for an interactive feedback loop with the environment [20]. Since the first breakthrough in the field of deep reinforcement learning [24], this field has seen considerable progress over the years, achieving super-human performance in computer games [23, 32, 31, 35], solving robotics control problem [2, 19, 11], improving recommendation systems [4, 13], healthcare [12], autonomous driving [25], and process optimization [27]. However, one of the limitations of online reinforcement learning is that it relies on interaction with a dynamical system or environment, and such online interactions can be expensive, especially in domains such as real-world robotics. Similar to supervised learning, where an algorithm trains a model by performing batch model updates, offline reinforcement learning performs model updates by sampling batches from a dataset of state, action, reward and next state (SARS) tuples. The model is then trained towards the standard reinforcement learning objective of maximizing expected future discounted reward, but also with a secondary objective of either keeping the learned policy close to the distribution of the *behavior policy* that was used to collect the data [9, 16].

We propose a method that allows to reduce the overall offline learning sample complexity by tracking model uncertainty measured from an ensemble of Q-networks. High level of uncertainty shows that the agent has entered an unexplored part of the state-action space indicating that introducing

36th Conference on Neural Information Processing Systems (NeurIPS 2022).

human demonstrations at this moment would have the most impact on learning. The algorithm measures the uncertainty and strategically introduces human demonstration data in-between the batches sampled from sub-optima non-expert data. Through our experiments in a simulated MuJoCo [34] and OffWorld Gym [15] environments, we demonstrate that our approach significantly reduces sample complexity compared to a model trained on a naively mixed dataset. The proposed method is directly transferable to the physical world without any additional assumptions on the properties of the environment or the learning algorithm.

## 2    Preliminaries

*Reinforcement learning* is a machine learning technique through which an agent learns to solve a task by learning from interactions with an environment. In the domain of robotic control and behavior most of the algorithms are based on the Markov Decision Process. A Markov Decision Process or MDP is defined as a tuple comprising of seven elements – $(\mathcal{S}, \mathcal{A}, \mathcal{T}, r, \gamma, \mathcal{S}_0, \mathcal{H})$, where $\mathcal{S}$ is the state space, $\mathcal{A}$ is the action space, $\mathcal{T}$ is the state transition probability function $\mathcal{T} = \mathcal{P}(s_{t+1}|s_t, a_t)$, $r$ is the environment reward function $r: \mathcal{S} \times \mathcal{A} \to \mathbb{R}^1$, $\gamma$ is the discount value, $\mathcal{S}_0$ is the start state distribution and $\mathcal{H}$ is the horizon length. In this work, we consider an MDP setting with a non-zero time horizon and discounted cumulative rewards. A reinforcement learning algorithm interacts with an environment to learn a policy $\pi$, which maximizes the reinforcement learning objective $\mathcal{J}(\pi)=\mathbb{E}[\sum_{t=0}^{H} \gamma^t r_t]$. The two important functions that help estimate the value of a state or a state and action pair are the *value function* $\mathcal{V}^\pi(s)=\mathbb{E}[\sum_{t=0}^{H} \gamma^t r_t|s_0=s]$ and the *Q-function* $\mathcal{Q}^\pi(s,a)=\mathbb{E}[\sum_{t=0}^{H} \gamma^t r_t|s_0=s,a_0=a]$. There are two broad categories of online reinforcement learning algorithms, viz. *on-policy* and *off-policy*. In an on-policy setting, the algorithm generates an action for a particular state based on the current policy. The recorded trajectory data is then used to update the current policy. Whereas in an off-policy setting there is a concept of a replay buffer $\mathcal{D}$, that accumulates data from different policies and the policy updates are made by sampling from the replay buffer instead of generating actions from the current policy. The standard Q-learning objective is to the minimize the Bellman error which is defined as $\mathcal{L}(\theta)=(\mathcal{Q}_\theta(s_t,a_t)-(r_t+\gamma \max_a \mathcal{Q}(s_{t+1},a)))$ [33].

In offline reinforcement learning, $\mathcal{D}$ represents the entire replay dataset and in Q-learning setting the objective is to minimize the following loss function based on Bellman error:

$$\mathcal{L}(\theta) = \mathbb{E}_{(s,a,r,s')\sim\mathcal{D}}[(\mathcal{Q}_\theta(s,a) - (r + \gamma \mathbb{E}_{a'\sim\pi(\cdot|s')}[\mathcal{Q}_{\theta'}(s', a')]))^2] \tag{1}$$

where $\theta'$ are the parameters of the target Q-network, softly updated for algorithm stability, while $\theta$ are the parameters of the running Q-network. In addition to the general reinforcement learning objective or the Q-learning objective, the algorithms usually have a secondary objective of keeping the learned policy close to the state-action distribution of the behavior policy. This is done to avoid overestimation of the Q-value, which happens when the Q-estimate error propagates during the bootstrapping, as shown in Equation 1. Over-estimation of Q-values can destabilize the training.

*Sub-optimal agent (SOA)* is represented by a behavior policy trained by QR-DQN learning algorithm to approximately 60% of optimal performance. While we assume that a human is an optimal (or close to) agent, the SOA represents a data source that has lower performance guarantees, but is easier to obtain.

## 3    Uncertainty-Guided Sampling in Offline RL

The problem we are addressing in this work is the reduction of sample complexity of existing offline RL algorithms. In robotics, solving a task with a minimal number of samples is a goal most practitioners would appreciate. In many cases, there is either an absence of an optimal approach or a human is an expert, but creating a human demonstration dataset is prohibitively expensive. Combining data from expert and non-expert demonstrations into a *mixed dataset* can significantly reduce the overall sample complexity [28], however in many practical applications the amount of human demonstrations necessary to reach close to optimal behavior remains prohibitively high.

In this work, we propose a method to strategically choose the order in which we pick samples from two replay buffers, one with data collected by a sub-optimal agent ($\mathcal{D}_{\text{SOA}}$) and another with data

collected from human demonstrations ($\mathcal{D}_\mathrm{H}$). Both are available ahead of time. This allows us to boost the speed of learning twice and reduce the overall sample complexity by 5 times, minimizing the required total sample size of both replay buffers required to reach close-to-optimal performance.

## 3.1 Sample complexity in offline RL setting

Offline RL algorithms learn a policy by performing policy updates on data sampled from an existing dataset. A dataset consists of trajectories collected by performing roll-outs in an environment using a behavior policy $\mu$. A dataset is usually of the form: $[(s_0^1, a_0^1, r_0^1, \ldots, s_\mathcal{H}^1, a_\mathcal{H}^1, r_\mathcal{H}^1), \ldots, (s_0^1, a_0^1, r_0^1, \ldots, s_\mathcal{H}^N, a_\mathcal{H}^N, r_\mathcal{H}^N)]$, where $N$ is the total number of episodes. A sample-efficient algorithm requires a smaller dataset than an algorithm with high sample complexity. Sample complexity bounds are usually computed based on MDP attributes such as the horizon length $\mathcal{H}$, the state space $\mathcal{S}$, the action space $\mathcal{A}$, etc. We will make the single policy concentrability assumption about the behavior policy as defined in [26, 37].

The *single policy concentrability* assumption is defined as follows: given a reference policy $\mu$ and an optimal policy $\pi^*$, $\mu$ is said to satisfy the assumption when

$$\max_{t \in [1..\mathcal{H}], (s,a) \in \mathcal{S} \times \mathcal{A}} \frac{d_t^{\pi^*}(s,a)}{d_t^\mu(s,a)} \leq C^*, \forall s \in \mathcal{S}, a \in \mathcal{A} \tag{2}$$

for some deterministic optimal policy $\pi^*$ and coefficient $C^*$, where $d$ is the state-action distribution of a policy. Intuitively the assumption requires there is a constant $C^*$ (concentrability coefficient) such that for every possible state-action pair the ratio of probabilities of said state-action being in $d^\pi$ and $d^\mu$ is not higher than the constant $C^*$.

The concentrability coefficient $C^* \in [1, \infty)$ is the smallest value that satisfies Equation 2. When the reference policy is exactly equal to the optimal policy $\mu = \pi^*$ the concentrability coefficient $C^* = 1$. $C^*$ estimates the difference between the state-action distribution density under of reference (behavior) policy and the optimal policy. Recent works in the field of sample complexity analysis of offline RL algorithms are based on the pessimism principle for value iterations [30, 37] and express the sample complexity of an offline RL algorithm as a function of $\mathcal{S}$, $\mathcal{A}$, $\mathcal{H}$ and $C^*$, where the upper bound on the number of episodes in the dataset $\mathcal{D}$ is $O(f(\mathcal{S}, \mathcal{A}, \mathcal{H}, C^*))$. The theoretical foundation of our approach relies on two assumptions. First, the algorithms need to satisfy the *single policy concentrability* assumption, so that their sample complexity could be expressed as a function of $\mathcal{S}$, $\mathcal{A}$, $\mathcal{H}$ and $C^*$. Second, we assume that the human demonstrator is an expert and the behavior policy $\mu_H$ corresponding to human policy is very close to an optimal policy $\pi^*$ such that the following condition is satisfied,

$$C_{\mu_H}^* < C_\mu^*, \forall \mu \in \mathcal{M} \tag{3}$$

where $\mathcal{M}$ is a family of non-expert behavior policies. Intuitively, inequality 3 states that if an algorithm's sample complexity can be expressed in terms of concentrability, and the assumption that human expert is close to optimal is satisfied, then the concentrability coefficient is lower when the algorithm is trained on human data.

## 3.2 Uncertainty estimation

In online reinforcement learning, uncertainty estimation has been leveraged for effective exploration strategies such as upper confidence bound exploration via Q-ensembles [5]. In an offline reinforcement learning setting, uncertainty estimation has been leveraged to act conservatively, and select paths with low uncertainty [3]. Our approach defines uncertainty in offline reinforcement learning as estimated variance of the Q-estimate of a given $(s, a)$ pair over an ensemble of $M$ Q-networks trained on the same data. In an ensemble of $M$ Q-networks that are trained to minimize the objective defined in Equation 1 we define point estimators for the mean $\mu = \frac{1}{M} \sum_{i=1}^{M} [Q_{\theta^i}(s,a)]$ and variance $\sigma^2 = \frac{1}{M} \sum_{i=1}^{M} (Q_{\theta^i}(s,a) - \mu(s,a))^2$ of the Q-estimates. The variance $\sigma^2$ characterizes the uncertainty in Q-function estimates and reflects the state of the training. At the start of the training, the Q-function estimates are expected to be noisy and have high variance. As the training progresses, the Q-network estimates become more accurate in the parts of the state space that were visited by the learning algorithm as the critic or Q-estimator improves.

### 3.3 Offline reinforcement learning with Uncertainty-Guided Expert Sampling

Introducing expert demonstrations into the training dataset helps reduce the sample complexity and also helps improve performance when compared to a mixed dataset. Having defined sample complexity analysis and uncertainty estimation in the previous sections, we propose the Uncertainty-Guided Expert Sampling (UGES) algorithm (see Supplementary Algorithm 1) that leads to more efficient use of available human data in conjunction with offline data collected from a sub-optimal behavior policy. Above-threshold uncertainty in the model ensemble triggers the algorithm to sample the next SARS tuple from the replay buffer $\mathcal{D}_H$ containing human demonstrations. Strategic sampling

---

**Algorithm 1** Offline Reinforcement Learning with Uncertainty-Guided Expert Sampling (UGES)

---

**Input:** Human dataset $\mathcal{D}_H = \left\{(s_t^i, a_t^i, r_t^i)\right\}_{i,t=1}^{N,H}$, Sub-optimal agent dataset $\mathcal{D}_{SOA} = \left\{(s_t^i, a_t^i, r_t^i)\right\}_{i,t=1}^{N,H}$, $E$ = Number of training iterations, $\epsilon$ = Uncertainty threshold, $M$ = Number of Q-networks

**Output:** $\mathcal{Q}_\theta$ = Q-function, for actor-critic methods only : $\pi_\phi$ = policy

1: **Initialization** Initialize Q-Ensemble $\mathcal{Q}_\theta^m = \mathcal{Q}_{\theta_0}^m \forall m \in [1..M]$, for actor-critic methods only: $\pi_\phi = \pi_{\phi_0}$, $\mathcal{S} = \{\}$ buffer for sampling

2: **for** training step $i$ in $\{1..E\}$ **do**

3:     Calculate $\mu = \frac{1}{M}\sum_{i=1}^{M}[Q_{\theta^i}(s,a)]$, $\sigma^2 = \frac{1}{M}\sum_{i=1}^{M}(Q_{\theta^i}(s,a) - \mu)^2$

4:     **if** $\sigma < \epsilon$ **then**

5:         $\mathcal{S}$ = Randomly sample from $\mathcal{D}_{SOA}$

6:     **else**

7:         $\mathcal{S}$ = Randomly sample from $\mathcal{D}_H$

8:     **end if**

9:     $\mathcal{L}(\theta)$ = Critic Loss based on an offline RL objective

10:     $\theta_t^i := \theta_{t-1}^i - \eta_Q \nabla \mathcal{L}(\theta) \forall \theta \in [1..M]$

11:     Policy improvement for actor-critic only: $\phi_t := \phi_{t-1} - \eta_\pi \mathbb{E}_{s\sim\mathcal{S},a\sim\pi_\phi(\cdot|s)}\left[\min_{1,M}\mathcal{Q}_i(s,a) - \log\pi_\phi(a|s)\right]$

12: **end for**

---

allows to keep the buffer of human demonstrations small while providing the same benefit to the overall learning process as a naive sampling on a larger human dataset would.

## 4 Experimental Results

We experimentally validate that uncertainty-based sampling strategy allows for a significant reduction in overall sample complexity and explore the benefits of combining the two sources of offline data. We compare two ways of mixing the data during learning: a naive (random) combination strategy is compared against the UGES method in terms of speed of learning, resulting agent performance, overall sample complexity, and amount of human data required to reach close-to-optimal performance.

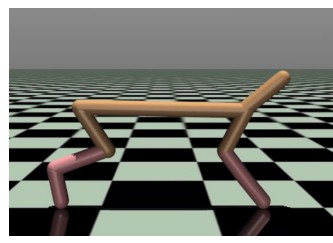
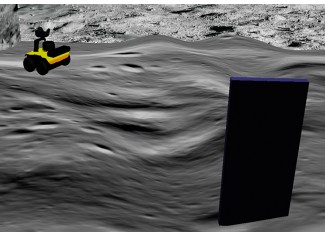

    (a) MuJoCo `Cheetah`         (b) MuJoCo `Ant`         (c) OffWorld Gym Monolith

Figure 1: Learning environments we used to test the uncertainty-guided expert sampling approach.

The experiments were performed using datasets generated from three learning tasks in simulated environments shown on Figure 1. In addition to the standard MuJoCo tasks `Cheetah` and `Ant` (D4RL

benchmark datasets [7]), UGES was tested in a simulated version of the OffWorld Gym `Monolith` environment that contains a vision-based task: a mobile robot explores the environment and gets rewarded when it reaches a monolith (goal) in the center of the field. A sparse reward of +1 is given when the robot is within a small distance from the goal with no step penalty.

To facilitate a fair comparison, we use *the number of successful trajectories* as the characteristic of a size of a dataset. A successful trajectory is defined as one that leads to a positive reward (see Supplementary Figure 3a for a comparison between training on human demonstrations (expert data) versus on data collected by a sub-optimal agent. In all experiment we maintain the 5:1 ratio of suboptimal to expert data. In `Cheetah` and `Ant` experiments the dataset was $320,000$ successful trajectories from the suboptimal agent and $80,000$ expert ones. In the `Monolith` environment the offline data consisted of $10,000$ SOA trajectories and $2,500$ expert human demonstrations. In our experiments, the uncertainty threshold, $\epsilon$, was tuned experimentally and the best threshold value was used in all three environments.

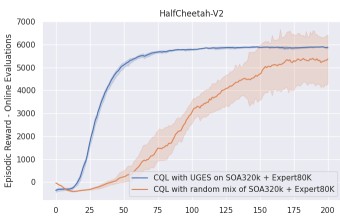 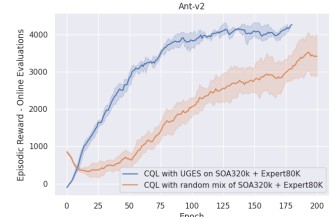 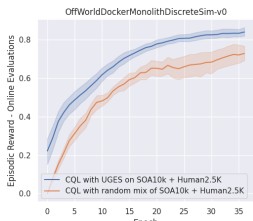

(a) Uncertainty-based sampling leads to more than 2x faster convergence in Mu-JoCO `Cheetah` environment and reaches better overall level of performance.

(b) Solving MuJoCo `Ant` environment with UGES from offline data is faster and leads to higher performance.

(c) In a simulation of a vision-based robotic task, UGES demonstrates more efficient data utilization and higher performance

Figure 2: Comparison between the episodic returns of CQL agents trained with UGES versus naive sampling of the human expert data. The confidence intervals are based on 5 to 10 runs.

## 4.1 Uncertainty-Guided Expert Sampling leads to a more efficient expert data utilization

Across all experiments we can see a significant improvement of data utilization efficiency when sampled are picked based on agent's uncertainty estimate. Subsequent experiments in the `Monolith` environment that allowed the naive mixing strategy to have access to a larger dataset of human expert trajectories showed that in this environment the naive strategy needed approximately 5x more successful human expert trajectories to follow UGES' learning curve (see Supplementary Figure 3b).

## 4.2 Uncertainty-Guided Expert Sampling reaches higher performance

We compare the episodic returns of online evaluations performed on agents trained using UGES versus Conservative Q-Learning (CQL) [17] agents trained on a combined dataset with 80% successful trajectories coming from the SOA agent and 20% being human expert trajectories. The result shown on Figure 2 demonstrates that already in the early stages of learning the algorithm proposed in this work makes a more efficient use of available samples, learning faster than a naive dataset combination strategy and consistently reaching higher level of performance.

## Conclusion

Generating a large dataset for offline RL can could be time-consuming, and algorithms that reduce sample requirements can make a difference. Human demonstrations provide great learning signal for Offline RL, but collecting such data is prohibitively expensive, especially in real-world robotics. These constraints lead us to consider using a combination of two sources of offline data: a limited amount of "expensive" human demonstrations with a dataset of "cheap" autonomously collected experiences. In this work we show how uncertainty estimation can guide strategic introduction of the human samples into the learning process leading to significant reduction in overall sample complexity.

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

# Supplementary Materials

## Tables and Figures

| Behavior policy | Successful trajectories | Time Steps |
|---|---|---|
| Human | 200 | 5,000 |
| Suboptimal agent | 300 | 13,500 |
| Suboptimal agent | 500 | 21,000 |
| Suboptimal agent | 900 | 40,000 |
| Suboptimal agent | 1,400 | 60,000 |
| Suboptimal agent | 1,600 | 70,000 |

Table 1: Correspondence between the number of successful trajectories and the total number of time steps in the dataset collected from the OffWorld Gym environment.

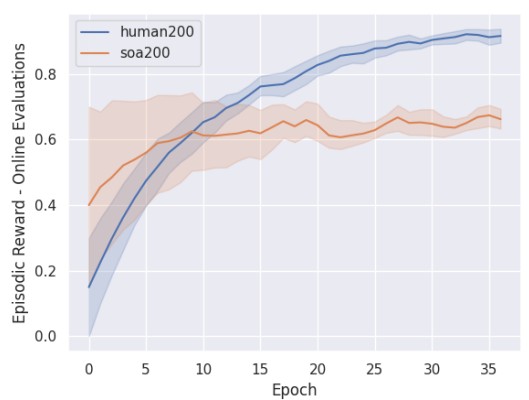

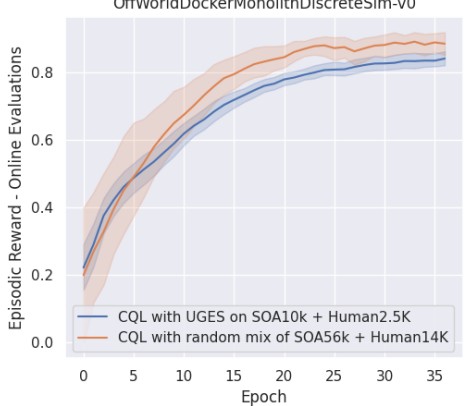

(a) With the same number of successful trajectories in the dataset, the trajectories provided by a human expert lead to faster learning and higher final performance.

(b) In OffWorld Gym Monolith experiment 5x human expert data were required for a naive sampling strategy to reaches a level of performance similar to UGES.

Figure 3: Supplementary experiments. **(a)** Comparing learning performance using human data versus relying on sub-optimal agent data only. **(b)** We conducted a set of experiment gradually increasing the amount of offline data available to the naive strategy to mark the moment when the performance reaches that of UGES.

## Related Work

Our work is most effective when applied with an algorithm that addresses some of the inherent issues of offline reinforcement learning like extrapolation error, q-value overestimation, and others. Extrapolation error is caused when there is a mismatch between the distribution of the dataset collected by a behavior agent and the state-action visitation of the policy being trained. Fujimoto et al. [9] introduced an algorithm called Batch-Constrained Q-Learning where they propose a solution to reduce the extrapolation error in reinforcement learning by training a policy such that it is constrained to stay close to the behavior policy. Techniques like BEAR-QL [16] prevent overestimation of q-value by bootstrapping error reduction, while CQL, and algorithm introduced by Kumar et al. [17], introduced a lower bound on the q-value to prevent over-estimation.

This work assumes that this algorithm is applied on a static dataset. To the best of our knowledge, the current state of the art algorithms like CQL and Implicit Q-Learning (IQL) [14] have been tested on static datasets only and there is no research work that demonstrates the application of offline reinforcement learning techniques on dynamic datasets.

Over the recent years, significant contributions have been made to the field of offline reinforcement learning towards solving such inherent challenges with the technique as deadly-triad issue [33], the issue with q-value function overestimation for out-of-distribution (OOD) state-action pairs [9], overfitting and underfitting issues [6], to name a few. To solve the problem of q-value function overestimation for OOD state-action pair, algorithms like conservative Q-learning (CQL) implement a learning objective that favors conservative estimates of the q-value function. The deadly-triad problem was solved in various works, viz. Fujimoto et al. [9], Lillicrap et al. [22] and Haarnoja et al. [11]. The formulation of overfitting and underfitting has been established in work by Kumar et al. [18], where several approaches to overcome these problems have been identified.

Since this work is based on mixing data generated by different behavior agents to create a single dataset, we rely on the results of Schweighofer et al. [28], Fu et al. [7], and Gulcehre et al. [10] that demonstrate the feasibility of training an offline RL model on a mixed dataset that mixes SARS tuples obtained by different behavior policies.

In the online RL setting, uncertainty estimation has been leveraged for efficient exploration [5], and for improving exploration by reducing the uncertainty of learned implicit reward function [21]. In the offline RL setting, the work by An et al. [1] quantifies the uncertainty of Q-value estimates by using an ensemble of Q-networks. However, in this work, uncertainty estimation is leveraged as a penalization term in Q-learning, while in our proposed method uncertainty acts as an indicator of the learning progress. Wu et al. [36] proposed the Uncertainty Weighted Actor-Critic algorithm that down-weights the contribution of OOD state-action pairs in training. This work achieves good performance gains on a dataset with narrow human demonstrations, however, in their work, data mixing is done by mixing the human demonstration data with data generated by imitating the human data. Some recent work [29], [8] have explored sampling strategies in offline RL using rank-based sampling or sampling prioritized experience replay where priority is given to sample transitions with lower epistemic uncertainty. Such methods may *over-sample* good samples.

## Limitations

The theoretical analysis in Section 3.1 is based on the assumption that the sample complexity of the CQL algorithm (and other offline RL algorithms that are being used in practice) can be expressed in terms of a set of MDP characteristics and the concentrability coefficient. However, to our knowledge, this assumption has not been explicitly proven for the CQL algorithm.

Due to a wide range of experimental conditions, we relied on data generated by interactions with a simulated environment, while our main aim is to make offline RL sample complexity sufficiently low to facilitate learning in the real physical world. The robotic benchmark learning environment chosen for this maintains a close relationship between its real and simulated versions of the environment. Since our results do not depend on the properties of the environment we believe that the empirical result shows in this work is directly transferable to the physical environment. Working with simulated data has allowed us to experiment with a broader set of experimental conditions than would not be possible otherwise. Our next immediate step is to validate the method on a real robot.

## Acknowledgements

The authors would like to thank Dylan Wishner and Felix Lu for advice and early version of the Tianshou code adaptation, and for suggestions and discussions they contributed during our work on this manuscript.

## Parameters

The table 2 contains the values of various parameters used in our MuJoCo experiments.

Parameters used in the Monolith environment are in table 3.

| Parameter | Value |
| --- | --- |
| Number of Ensemble, M | 10 |
| Uncertainty threshold, $\epsilon$ | 16 |
| Actor learning rate, $\eta_\pi$ | 0.0001 |
| Critic learning rate, $\eta_Q$ | 0.0003 |
| Batch Size | 256 |
| Number of iterations, $E$ | 36 |
| Gamma, $\gamma$ | 0.99 |
| RL Algorithm | Soft-Actor Critic |
| Hidden layer sizes | [256, 256] |

Table 2: Values of various parameters used in experiments.

| Parameter | Value |
| --- | --- |
| Number of Ensemble, M | 10 |
| Uncertainty threshold, $\epsilon$ | 16 |
| Critic learning rate, $\eta_Q$ | 0.0001 |
| Batch Size | 32 |
| Number of iterations, $E$ | 36 |
| Gamma, $\gamma$ | 0.99 |
| RL Algorithm | QR-DQN |

Table 3: Values of various parameters used in experiments.

