# OpenReview forum: "Offline Robot Reinforcement Learning with Uncertainty-Guided Human Expert Sampling"
_NeurIPS.cc/2022/Workshop/Offline_RL — Offline RL Workshop NeurIPS 2022_

### Official Review · Reviewer_7oVS · 2022-10-18
**Interesting idea, though need more intuition for why it works**

**Rating:** 7
**Confidence:** 4

**Review:**

The authors present a technique for offline RL that strategically mixes sub-optimal data and expert data based on a measure of uncertainty. When the agent's uncertainty about a transition is high, the model is trained on expert data, otherwise its trained on sub-optimal data. Experiments show this mixing strategy leads to faster convergence and higher performance with less expert data than naive mixing.

The explanation of the algorithm could benefit from more detail, particularly more written explanation. I couldn't figure out all the details from the pseudo-code. My understanding is that at each step the uncertainty is estimated for a random batch of transitions. Then if the uncertainty is high, the next batch is chosen from the expert data, otherwise it's chosen from the sub-optimal data.
However, I'm not sure how this leads to the benefits shown in the experiments. I don't have an intution for what this accomplishes and why just mixing expert and non-expert data in each batch wouldn't work just as well.

Pros
- Experiment results show clear benefit of the method.
- Reducing the number of expert demos required to get the same performance is an important problem.

Cons
- Little explanation of why the method works.

Small things
- > This approach allows us to boost the speed of learning by Nx and reduce the overall sample complexity by Mx, minimizing
the required total sample size of both replay buffers required to reach close-to-optimal performance.

Should there be actual numbers here?
- It would be helpful in the experiments to list both number of successful trajectories and the total size of the dataset (maybe just report dataset size with percent successful trajectories).

---

### Official Review · Reviewer_atHQ · 2022-10-21
**Interesting**

**Rating:** 6
**Confidence:** 2

**Review:**

This paper considers learning offline RL policies in a setting where the agent is allowed to ask for "more data". An algorithm is proposed that uses variance of an ensemble of CQL value functions to decide whether or not to query for more data.

I think this area of research is very interesting, and to the best of my knowledge, not well-studied.

I think the experiments can be done in a more focused manner, though. It seems that the utility of an algorithm here is not necessarily how many less gradient updates it takes to learn a good policy, but rather how many fewer expert trajectories are necessary. I recommend the authors for future iterations to refocus the experimental section to study how performance varies depending on the number of additional trajectories injected (and how the author's adaptive selection strategies compares to these baselines).